# HIV–HPV Co-Infection and Identification of Novel High-Risk HPV Among Women at Two Hospital Centers in Cotonou, Republic of Benin

**DOI:** 10.3390/v17050714

**Published:** 2025-05-16

**Authors:** Clémence D. Gouton, Ifeoluwa O. Bejide, Oludayo O. Ope-ewe, Marius Adjagba, Simon Azonbakin, Gaonyadiwe Muzanywa, Florence T. Akinyi, Arnaud Agbanlinsou, Yanique Goussanou, Onikepe Folarin, Anatole Laleye, Christian T. Happi, Chinedu A. Ugwu

**Affiliations:** 1The Africa Centre of Excellence for Genomics of Infectious Diseases (ACEGID), Redeemer’s University Nigeria, Ede PMB 230, Nigeria; clemencegouton@gmail.com (C.D.G.); bejidei@run.edu.ng (I.O.B.); ope-eweo@run.edu.ng (O.O.O.-e.); gmuzanywa@gmail.com (G.M.); florencetaduda@gmail.com (F.T.A.); folarino@run.edu.ng (O.F.); happic@run.edu.ng (C.T.H.); 2Department of Biological Sciences, Faculty of Natural Sciences, Redeemer’s University, Ede PMB 230, Nigeria; 3Laboratory of Histology-Reproductive Biology, Cytogenetics and Medical Genetics, Human Biology Unit, Faculty of Health Sciences, Abomey-Calavi University, Cotonou 01 BP 188, Benin; mariusoadjagba@yahoo.fr (M.A.); azandeg@yahoo.fr (S.A.); arnaudagbanlinsou@yahoo.fr (A.A.); ygyanne@outlook.fr (Y.G.); cytogenbenin@gmail.com (A.L.)

**Keywords:** human papillomavirus (HPV), HIV, cervical cancer, HPV–HIV co-infection, risk factors, prevalence, Sanger sequencing

## Abstract

Persistent high-risk human papillomaviruses (HR-HPVs) infection is the leading cause of cervical cancer. With over 200 circulating genotypes, HPV detection, management, and prevention remain challenging. In Benin, HPV prevalence and genotype distribution are largely unknown, and no national HPV vaccination program exists. This study investigates the prevalence, genotypic diversity, and risk factors of HIV–HPV co-infection among women in Cotonou, Benin. Cervical swabs were collected from 100 women living with HIV (WLWHIV) and 51 women without HIV (WWHIV) at two hospitals. DNA extraction and nested polymerase chain reaction (PCR) were used to detect HPV, followed by Sanger sequencing for genotyping. Chi-squared analysis was used to assess risk factors. HPV was detected in 85% (85/100) of WLWHIV and 60.8% (31/51) of WWHIV (*p* = 0.002), confirming HIV as an independent risk factor. Fifteen HR-HPV genotypes were identified, with HPV 45 most prevalent in WLWHIV and HPV 16 in WWHIV. Notably, HR-HPV 67, 70, and 82 were detected for the first time in Benin. Unmarried status and detectable HIV load were significant risk factors for co-infection. The high HPV prevalence, particularly among WLWHIV, underscores the urgent need for HPV surveillance and vaccination in Benin. Identifying novel HR-HPV genotypes highlights the necessity for ongoing monitoring and targeted prevention strategies.

## 1. Introduction

Human papillomaviruses (HPVs) are among the most prevalent sexually transmitted infections worldwide, affecting both men and women [1,2]. With over 200 identified genotypes [3,4], HPVs are classified as high-risk (HR-HPV) or low-risk (LR-HPV) based on their oncogenic potential [5,6]. Persistent HR-HPV infection is the primary cause of cervical cancer (CC) in women [7]. In the majority of cases, cervical cancer occurs when the HPV genome integrates into the host genome. During HPV DNA integration, the viral genome usually breaks in the E1/E2 region, leading to the increased expression level of E6 and E7 proteins, the most oncogenic macromolecules ever discovered [8]. E6 from the carcinogenic HPV types inhibits the onco-suppressive functions of p53, thereby preventing cell cycle arrest or apoptosis induction [9,10]. E7 promotes the degradation of pRb, leading to aberrant activity of E2F transcription factors that promote S-phase re-entry of differentiating cells to provide a replication-competent environment. This dysregulation causes cell abnormalities or precancerous lesions in the basal cell layer, which may result in cervical cancer [8]. While most infections resolve within two years, immunocompromised individuals, particularly those living with HIV, are more likely to experience persistent infections, increasing their risk of HPV-related malignancies [11,12,13].

HPV prevalence varies widely across Sub-Saharan Africa (SSA), ranging from 10.7% in Ghana to 90.8% in Côte d’Ivoire [14]. The genotypic distribution also differs by region, with HPV 52 and 31 predominant in Senegal, HPV 35 and 16 in Nigeria, and HPV 52 and 59 in Burkina Faso [14]. However, data on HPV prevalence and circulating genotypes in the Republic of Benin remain limited, particularly among high-risk populations [15,16]. A study by Capo-Chichi et al. (2016) reported a higher HR-HPV prevalence among women living with HIV (WLWHIV) in Benin (31%) compared to HIV-negative women (23%), with WLWHIV also more likely to have multiple HR-HPV infections [15]. This increased susceptibility places WLWHIV at a greater risk of developing HPV-related precancerous lesions, which tend to progress more rapidly and respond less effectively to treatment in immunocompromised individuals [17,18].

Vaccination is a key strategy for HPV prevention. Six HPV vaccines—bivalent (Cervarix, Cecolin, and Walrinvax), quadrivalent (Gardasil and Cervarix), and nonavalent (Gardasil 9)—are currently licensed for use by the WHO [19]. The quadrivalent ones protect against HR-HPV types 16 and 18, as well as LR-HPV types 6 and 11 [20]. The bivalent HPV vaccines target HPV-16 and HPV-18 [20,21], while Gardasil 9 extends protection to five additional oncogenic types (HPV-31, HPV-33, HPV-45, HPV-52, and HPV-58) [20,21]. Although HPV vaccination programs have expanded across SSA, Benin has yet to introduce any HPV vaccines. While vaccination has contributed to a decline in HPV infections globally, circulating genotypes vary across populations and over time due to geographic and health-related factors [14]. This highlights the need for continuous surveillance to determine the most effective vaccine strategy for the Beninese population.

This study investigates the prevalence, genotypic diversity, and risk factors associated with HIV–HPV co-infection at two hospital centers in Cotonou, Republic of Benin. We hypothesize that WLWHIV have high HPV prevalence and highly diverse HR-HPV genotypes compared to WWHIV. Findings from this study will provide critical epidemiological data to support policy decisions, enhance disease management, and strengthen public health interventions for HPV prevention and control.

## 2. Methods

### 2.1. Study Design and Population

This was a cross-sectional study conducted in two University Hospitals in Cotonou, Republic of Benin (Mother and Child University Hospital Center of Lagune (CHU-MEL) and National University Hospital Center Hubert Koutoukou Maga (CNHU-HKM)) between November 2023 and January 2024. Enrolled participants were sexually active WLWHIV (100) and WWHIV (51), aged 18 years and above, who had provided informed consent freely. WLWHIV attended healthcare facilities to receive their antiretroviral therapy (ART). After signing the consent form, a questionnaire was administered to participants to collect information about sociodemographics, sexual behavior, and HIV history. In WLWHIV, recent CD4+T lymphocyte cell count (within 2 weeks before sample collection) and HIV load (within one month before sample collection) were retrieved from their hospital medical records. Information on prior cytological examination of the cervicovaginal smear was also collected from participants.

The sample size was determined using the equation described by Naing et al. [22]:$$n=\frac{Z^{2}P(1-P)}{d^{2}}$$

*n* is the minimum sample size, Z is the statistical score corresponding to our confidence level, P is the expected prevalence of HPV among WLWHIV in Cotonou, and d is the margin of error.

The expected prevalence of HPV was 31% among WLWHIV in Cotonou [15].$$\begin{array}{l} n=\frac{1.96^{2}(0.31)(1-0.31)}{{\left(0.1\right)}^{2}}\\ \approx 83 \end{array}$$

Therefore, the minimum sample size estimated for this study was 83 WLWHIV. For comparison, 42 samples from WWHIV (such that the ratio between WLWHIV and WWHIV cohorts is 2:1) were to be enrolled alongside the WLWHIV cohort. To increase representativeness, a total of 100 WLWHIV and 51 WWHIV were enrolled in this study.

### 2.2. Study Procedure

Cervical swabs for HPV detection were collected by trained midwives using a disposable sterile speculum, which was introduced in the vagina along with a disposable cytobrush to reach the cervical–uterine junction, where cells were collected by rotating the cytobrush clockwise five times. The brush was placed in a 50 mL collection tube containing 10 mL of sterile 1× phosphate-buffered saline (1×PBS) to collect cells [15]. Cells were kept on ice and delivered to the laboratory within four to five hours. All the samples were stored at −20 degrees Celsius upon arrival at the Laboratory of Cytogenetic and Molecular Biology at the Faculty of Health Sciences of Cotonou, Benin Republic.

### 2.3. DNA Extraction

Prior to extraction, the cervical swab samples were thawed at room temperature for 15 min and then transferred to 15 mL falcon tubes for centrifugation at 3000× *g* for 10 min at 4 °C. After centrifugation, the supernatant was discarded, and the pellet was washed in 2 mL of 1×PBS, followed by centrifugation at 3000× *g* for 10 min at 4 °C. The cell pellet was resuspended in 500 µL of PBS [23]. Using the Qiagen QIAmp DNA mini kit, 200 µL of the resuspended cell pellet was used for DNA extraction according to the manufacturer’s instructions (QIAGEN, Hilden, Germany).

### 2.4. ß-Globin Amplification

After extraction, the quality of the DNA was assessed by amplifying the ß-globin gene using PCO3/PCO4 primer sets to ensure that the DNA was not degraded before setting up the HPV PCR assay (Appendix A).

The ß-globin PCR reactions were carried out in 20 µL total volume consisting of 4 µL master mix (5X FIREPol^®^ Master Mix), 1 µL of PCO3, 1 µL of PCO4, 9 µL of nuclease-free water and 5 µL of genomic DNA. The reaction mixtures were subsequently subjected to thermal cycling (Eppendorf, Hamburg, Germany) with the following conditions: initial denaturation at 95 °C (5 min), 40 cycles of denaturation at 94 °C (1 min), annealing at 55 °C (1 min), and extension at 72 °C (1 min), concluding with a final extension at 72 °C (10 min).

### 2.5. HPV L1 Gene Amplification

A nested PCR was performed to detect HPV DNA in the 151 extracted samples, using three primer sets targeting the HPV L1. The first round of PCR involved a multiplex assay using MY09/11 and PGMY11A/09F primers to amplify a 450 bp region of the HPV L1 gene. The second round used GP5+/GP6+ primers to optimize the detection of a 150 bp product (Appendix A).

The primary amplification used a 20 µL reaction containing 4 µL FIREPol master mix (solis bio dyne, Tartu, Estonia), 7 µL nuclease-free water, 1 µL of each primer, and 5 µL genomic DNA. Thermal cycling included initial denaturation at 95 °C (5 min), followed by 40 cycles of denaturation at 94 °C (1 min), annealing at 55 °C (1 min), and extension at 72 °C (1 min), concluding with a final extension at 72 °C (5 min).

The second nested amplification was conducted in 20 µL total volume consisting of three microliters (3 µL) of the amplicon from the first round, used as DNA template, 4 µL FIREPol master mix, 1 µL of forward primer, 1 µL of reverse primer, and 11 µL of nuclease-free water. The reaction mixtures were subjected to similar cycling conditions, except the annealing temperature at 42 °C.

### 2.6. Gel Electrophoresis

After the globin gene PCR and the second round PCR, gel electrophoresis was conducted to observe the expected band size (110 bp for the globin gene and 150 bp for the second round of HPV PCR). PCR products, along with a 100 bp DNA ladder (New England Biolabs, Ipswich MA, USA), were loaded onto a 2% agarose gel. Electrophoresis was conducted at 120 V, 150 mA, and 100 W for 35 min using the Bio-Rad PowerPac 300 power supply (Bio-Rad Laboratories, Inc., Berkeley, CA, USA) (Appendix A).

### 2.7. Sanger Sequencing

Due to the availability of resources, 90 samples out of the 116 samples in which HPV was detected were subjected to Sanger sequencing to determine their HPV genotypes. The 90 samples were selected based on the quality of the bands observed during gel electrophoresis, with priority given to samples exhibiting strong, well-defined bands. This approach ensured optimal DNA quality and concentration for successful downstream sequencing and data reliability. Using the BigDye™ Terminator v3.1 cycle sequencing kit (Thermo Fisher Scientific, Foster, CA, USA) on the Genetic analyzer 3500 (Thermo Fisher Scientific, Foster, CA, USA), the PCR products (150 bp amplicon) of the second round of PCR were purified and subjected to sequencing. The amplicon encompassed the hypervariable region of the L1 gene.

### 2.8. Bioinformatic Analyses

The chromatograms obtained from sequencing were first trimmed to remove background noise using BioEdit Sequence Alignment Editor Version 7.0.5.3 (10/28/05). The chromatogram of the reverse DNA strand was then converted into nucleotide text format using Microsoft Word Office Professional Plus 2016. These reverse sequences were used to generate reverse complements with APE (v3.1.5). Both the reverse sequences and their complements were assembled de novo into contigs online using EMBOSS Merger. The resulting contigs were analyzed using BLAST (BLAST+ version 2.16.0) to identify HPV types based on sequence similarity, which ranged from 79.9% to 90% [24].

To construct phylogenetic trees for novel HR-HPVs found in this study (HPV67, HPV 70, and HPV 82), contig sequences were aligned with reference sequences downloaded from NCBI GenBank using MAFFT v7 [25] and MEGA 5.2 [26]. The aligned sequences were trimmed in MEGA 5.2 to remove overlapping or poorly aligned regions. Phylogenetic trees were constructed in MEGA version 11.013 [27] using the maximum likelihood method and general time-reversible model [28]. The bootstrap consensus tree inferred from 1000 replicates [29] is taken to represent the evolutionary history of the taxa analyzed. Branches corresponding to partitions reproduced in less than 50% of bootstrap replicates are collapsed. Initial tree(s) for the heuristic search were obtained automatically by applying Neighbor-Join and BioNJ algorithms to a matrix of pairwise distances estimated using the maximum composite likelihood (MCL) approach and then selecting the topology with superior log likelihood value. Codon positions included were 1st+2nd+3rd+Noncoding. Evolutionary analyses were conducted in MEGA11 [27].

The region of the L1 gene used for alignment for each HPV type is shown in Appendix A.

### 2.9. Statistical Analyses

Data were processed and analyzed using SPSS software version 20. Frequencies were calculated to describe the data and assess HPV prevalence and genotype distribution. Chi-squared tests were used to explore associations between HPV infection and HIV status, age, marital status, number of sexual partners, age at first sexual intercourse, CD4+ counts, and HIV viral load. Statistical significance (*p* ≤ 0.05) was reported with odds ratios (ORs) and 95% confidence intervals (CIs).

## 3. Results

A total of 151 females participated in this study, including 100 women living with HIV (WLWHIV) (66.2%) and 51 HIV-negative women (WWHIV) (33.8%). All WLWHIV were on antiretroviral therapy (ART), with 52% (52/100) having CD4+ counts >500 cells/mm^3^ and 42% (42/100) with undetectable viral loads. The mean age (±SD) of participants was 41.78 ± 11.9 years, with WLWHIV significantly older than WWHIV (43.5 ± 10.7 vs. 39.9 ± 11.2 years, *p* = 0.033). The median age at first sexual intercourse was 18 years (IQR: 12–27), and the median number of lifetime sexual partners was 2 (IQR: 1–10). The majority of participants were either married (50.9%, 77/151) or single (35.1%, 53/151), and 74% reported initiating sexual activity between the ages of 15 and 20 (Table 1).

Table 1 presents the demographic differences between WLWHIV and WWHIV. Several factors, including marital status, education level, number of sexual partners, age at first sexual intercourse, and alcohol consumption, showed statistically significant differences between the two groups. Most WLWHIV had an undetectable viral load and CD4+ counts above 500.

HPV infection was detected in 116 participants (76.8%), with a significantly higher prevalence in WLWHIV (85%, 85/100) compared to WWHIV (60.8%, 31/51) (*p* = 0.002). Among participants with previously negative cytology results, 64.3% (9/14) tested positive for high-risk HPV (HR-HPV). HPV positivity was confirmed based on band sizes at 150 bp (Appendix A).

Among WLWHIV, detectable HIV viral load and unmarried status were significant risk factors for HPV infection. Unmarried WLWHIV had nearly five times the odds of testing positive for HPV compared to married individuals, while those with detectable viral loads were five times more likely to test positive for HPV compared to those with suppressed viral loads. Other factors such as age, oral contraceptive use, inconsistent condom use, number of sexual partners, age at first intercourse, ART duration, and CD4+ count showed no significant association with HPV infection (Table 2).

Table 2 displays the risk factors of HPV–HIV co-infection. Among the factors assessed, only unmarried women (single, divorced) and women with detectable HIV load were at high risk of HPV–HIV co-infection. Unmarried WLWHIV had nearly five times the odds of testing positive for HPV compared to married individuals. At the same time, those with detectable viral loads were five times more likely to test positive for HPV compared to those with suppressed viral loads. The remaining factors, such as age, oral contraceptive use, inconsistent condom use, number of sexual partners, age at first intercourse, ART duration, and CD4+ count, showed no significant association with HPV infection.

Of the 116 HPV-positive samples, 90 were selected for Sanger sequencing based on band quality, and 67 samples (74.4%) yielded clean sequence data, including 46 from WLWHIV and 21 from WWHIV (Appendix A). The sequences were deposited in GenBank under accession numbers PQ868536–PQ868556.

In total, 26 distinct HPV genotypes were identified, with HR-HPVs accounting for 56.7% (38/67) and LR-HPVs for 43.3% (29/67). Fifteen HR-HPV types were identified, including HPV 16, 18, 33, 35, 39, 45, 52, 53, 56, 58, 66, 67, 68, 70, and 82 (Table 3), while 11 LR-HPV types were identified, including HPV 6, 32, 42, 54, 61, 62, 72, 81, 83, 90, and 114 (Table 3).

Among HPV-positive WLWHIV, 56.7% tested positive for HR-HPVs, with HPV 45 being the most prevalent (19.2%) (Figure 1a). LR-HPVs were found in 43.5%, with HPV 62 and HPV 81 being the most frequent (20%). In WWHIV, 57.1% tested positive for HR-HPVs, with HPV 16 being the most common (25%) (Figure 1b). LR-HPVs were present in 42.9%, with HPV 54 being the most frequent (33.3%). HR-HPV prevalence was similar between WLWHIV (56.5%) and WWHIV (57.1%) (OR = 1.02, *p* = 0.96).

Table 3 shows the frequency of HR-HPV in both WLWHIV and WWHIV. HPV 33 and HPV 45 were the most prevalent HR-HPV genotypes. HPV-18, 39, 45, 56, 58, and 82 were exclusive to WLWHIV, while HPV-35, 67, and 68 were found only in WWHIV.

Phylogenetic analysis was performed to compare newly identified HR-HPV genotypes (HPV 67, 70, and 82) in Benin with reference sequences from the NCBI database (ID: 2893894). HPV 82 was exclusive to WLWHIV, while HPV 67 was found only in WWHIV. HPV 70 was detected in both groups. The HPV 67 sequence was closely related to a 1997 Italian sequence (Y12207.1, Figure 2a). HPV 70 sequences clustered separately, indicating a recent common ancestor distinct from NCBI sequences (Figure 2b). The HPV 82 sequences grouped with a 2013 New York sequence (KF436803.1, Figure 2c), suggesting evolutionary links.

## 4. Discussion

This study reveals a high prevalence of HPV in both WLWHIV (85%) and WWHIV (60.8%), with a significantly higher rate among WLWHIV (*p* = 0.002). Detectable HIV viral load and single marital status were identified as key risk factors for HPV detection in WLWHIV. Notably, HPV 45 was the most prevalent HR-HPV type in WLWHIV, whereas HPV 16 was the most common in WWHIV.

The overall HPV prevalence of 76.8% observed in this study is markedly higher than the 33.2% reported by Piras et al. in Benin in 2011 [16]. This discrepancy may stem from differences in study populations, as Piras et al. examined the general population across various regions of Benin without specifically targeting HIV-infected individuals. The significantly higher prevalence of HPV among WLWHIV compared to WWHIV supports the well-established association between HIV and increased susceptibility to HPV infection. This finding aligns with prior research demonstrating that HIV-infected individuals have a higher likelihood of acquiring and maintaining persistent HPV infections due to immunosuppression [12,31,32]. While biological mechanisms suggest a bidirectional relationship between HIV and HPV, further studies are needed to elucidate the exact nature of this interaction [32].

The 85% HPV prevalence among WLWHIV in this study is higher than rates reported in other African countries, including Ghana (60%) [33], Nigeria (28%) [34], and Ethiopia (30.4%) [35].

Similarly, the 60.8% HPV prevalence among WWHIV in this study is higher than rates reported in studies from South-West Tanzania (36%) [36], Senegal (18%) [37], and a review conducted in sub-Saharan Africa (26.5%) [38]. These variations may be attributed to differences in sample sizes, detection methodologies, and population characteristics, including sexual behaviors. The use of consensus primers in this study, which detects a broader spectrum of HPV types, may have also contributed to the higher prevalence. The lower HPV prevalence reported in studies from Nigeria, Ethiopia, Senegal, and Tanzania could also be attributed to the fact that these countries have implemented national HPV vaccination programs, which likely prevented some HPV infections compared to Benin, where such a program has not been implemented [39,40].

A particularly concerning finding was the high prevalence (64.3%) of HR-HPV among women with previously negative cytological screening results. This underscores the limitations of relying solely on cytology-based screening and emphasizes the need for routine HPV testing as a complementary method for early detection of individuals at risk of cervical cancer [41,42]. However, it is also possible that these women acquired HPV infections after their initial screening.

Detectable HIV viral load and single marital status emerged as significant risk factors for HPV infection among WLWHIV. Women with detectable HIV viral load were five times more likely to test positive for HPV, likely due to impaired immune responses that facilitate persistent infections. Similarly, unmarried women were nearly five times more likely to have HPV, possibly due to a higher number of sexual partners and increased exposure to the virus. Consistent with our findings, Qiao et al. (2020) reported an association between detectable HIV viral load and HPV infections in China [43]. However, studies from South Africa [44] and Italy [32] found no correlation between HIV viral load and HPV infection, suggesting that additional factors, such as behavioral differences and unmeasured confounders, may influence HPV acquisition.

The association between single marital status and HPV infection aligns with previous studies in Italy and Pakistan [32,45,46]. Similarly, the observed relationship between HPV prevalence and HIV viral load, independent of CD4+ cell counts, is consistent with findings indicating that a higher HIV load is linked to HR-HPV persistence, irrespective of CD4+ T-cell levels [47,48]. This suggests that uncontrolled HIV infection may compromise immune defenses, allowing HR-HPV infections to persist, which would otherwise be naturally cleared in WWHIV [49].

In both WLWHIV (56.5%) and WWHIV (57.1%), HR-HPV was more prevalent than LR-HPV, corroborating findings from Piras et al. (2011) in Benin, which reported an HR-HPV prevalence of 88% in the general population [16]. However, our data contrasted with Capo-Chichi et al. (2016) in Benin, which reported significantly lower HR-HPV rates in both WLWHIV (31%) and WWHIV (23%) [15]. The difference may be attributed to variations in detection methods, as the present study identified only single infections, whereas Capo-Chichi et al. included multiple infections in their analysis.

HPV 45 and HPV 16 were the predominant HR-HPV types in WLWHIV and WWHIV, respectively, differing from the Capo-Chichi et al. (2016) study [15], which found HPV 56 and HPV 66 to be the most common types. Notably, five of the 12 HR-HPV genotypes identified in WLWHIV in this study—HPV 16, 18, 53, 70, and 82—were absent in the previous study, while four of the previously reported types—HPV 42, 43, 35, and 31—were not detected here. This suggests temporal shifts in HPV genotype prevalence, possibly due to changing population immunity, further emphasizing the need for continuous HPV surveillance in Benin.

In WWHIV, the most prevalent HR-HPVs were HPV 16 (25%), followed by HPV 66 (16.7%). A review reported HPV-16 (6.7%) and HPV 66 (6.8%) as the most prevalent HR-HPVs in HIV-negative women from Sub-Saharan Africa [38]. Similarly, a study from Senegal reported HPV 16 as the most common HR-HPV, with a prevalence of 2.4% [37]. The smaller sample size may partly explain the high prevalence observed in our study. The size of WWHIV in this study is much less than that of the compared studies. Interestingly, the second most frequent HR-HPV in this study (HPV 66) was found as the first most frequent HR-HPV in the study conducted in Benin by Capo-Chichi et al. [15]. In contrast to our finding, a study from Nigeria reported HPV 82 as the most prevalent HR-HPV, followed by HPV 16 and 18 among HIV-negative women [50]. Surprisingly, HPV 18 was not found in our HIV-negative cohort despite being frequently reported in previous studies [15,37,38,50].

Looking at the phylogenetic tree, sequences of HPV 70 from this study form distinct clades, indicating that they share a common ancestor that is genetically distant from the reference HPV 70 sequences downloaded from NCBI. Notably, two sequences from WLWHIV in this study (EMBOSS GCP 051 et EMBOSS GCP 058) share 100% similarity, suggesting a direct transmission pattern between individuals. These two sequences are also closely related to the other two HPV 70 sequences from this study: EMBOSS GCP083 (WLWHIV) and EMBOSS GCNM049 (WWHIV) (EMBOSS GCP083 et EMBOSS GCNM049). Interestingly, HPV 70 sequences from WLWHIV show higher genetic similarity among themselves compared to those from a WWHIV. However, all the HPV 70 sequences from this study remain genetically distant from the reference sequences in GenBank. Sequences of HPV 67 (EMBOSS GCNM035) and HPV 82 are closely related to HPV sequences from Italy (1997) [51] and New York, USA (2013) [30], respectively, suggesting the HPV genotypes may have been imported to Benin. This may be due to migration or international travel, suggesting that the women in this study may have interacted with individuals from that country.

A key contribution of this study is the identification of novel HR-HPV types (HPV 67, 70, and 82) and LR-HPV types (HPV 90 and 114) in Benin for the first time. The discovery of these HPV genotypes may be attributed to the use of broader-spectrum primers, unlike previous studies that employed type-specific assays [15,16,52]. Interestingly, these novel HR-HPV types have been reported in other African countries. In Tanzania, HPV 67 and 70 were reported in both HIV+ and HIV- women, while HPV 82 was only detected in HIV-negative women [36]. HPV 70, 82, and 90 were reported in Women with or without cervical cancer in Nigeria [50,53]. HPV 82 was also reported in Senegal [37]. Cross-protective immunity among HPV types has been reported; however, this has not been shown for the novel HR-HPV types identified in this paper [54,55]. Currently, no available HPV vaccines provide protection against these newly identified HR-HPVs, raising concerns about vaccine coverage in Benin and other African countries. Given the absence of an HPV vaccination program in the country, these findings underscore the urgent need for further epidemiological studies to assess the prevalence and clinical relevance of these novel HPV types.

The high prevalence of HPV 45 in WLWHIV suggests that the currently available bivalent and quadrivalent vaccines may offer limited protection in this population, as they do not provide full protection against the most frequently detected HR-HPV genotypes, compared to the nonavalent Gardasil 9 vaccine [55,56]. Therefore, the nonavalent vaccine, which provides the greatest protection against HR-HPV 45 and 33, should be considered for national implementation.

Overall, this study underscores the substantial burden and diversity of HPV genotypes in Benin, particularly among WLWHIV, where HR-HPV infections predominate. Given the asymptomatic nature of HPV and the low cervical cancer screening rates in Africa [36], these findings reinforce the need for a national HPV screening and vaccination program. Expanding national HPV surveillance to include both WLWHIV and WWHIV is crucial to enable early detection and intervention. Furthermore, the introduction of the nonavalent Gardasil 9 vaccine should be prioritized to ensure broader coverage against the most prevalent HR-HPV genotypes in the country.

## Figures and Tables

**Figure 1 viruses-17-00714-f001:**
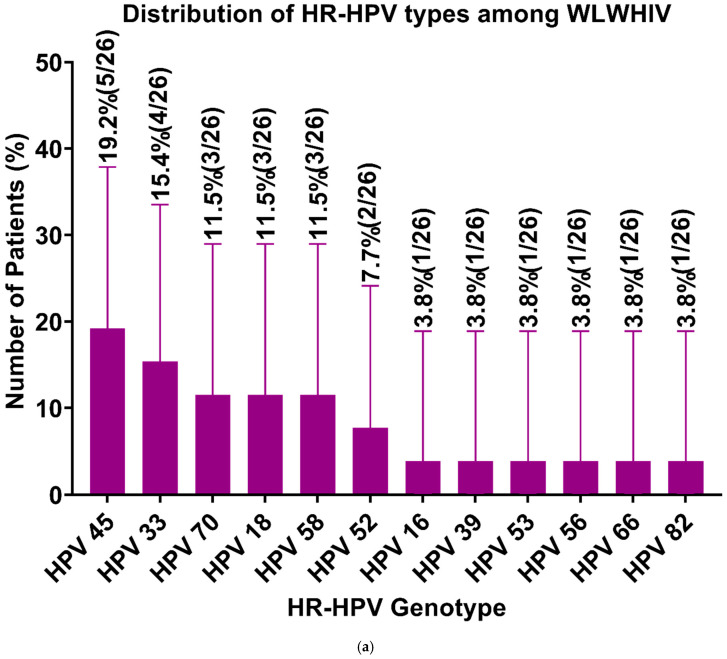
Distribution of HR-HPV among WLWHIV and WWHIV. (**a**) shows that HPV 45 was the most prevalent HR-HPV type (19.2%; 5/26) among WLWHIV. (**b**) shows that HPV 16 was the most prevalent HR-HPV type among WWHIV (25%; 3/12). The number of participants in percentage for each HR-HPV type was calculated at a 95% confidence interval.

**Figure 2 viruses-17-00714-f002:**
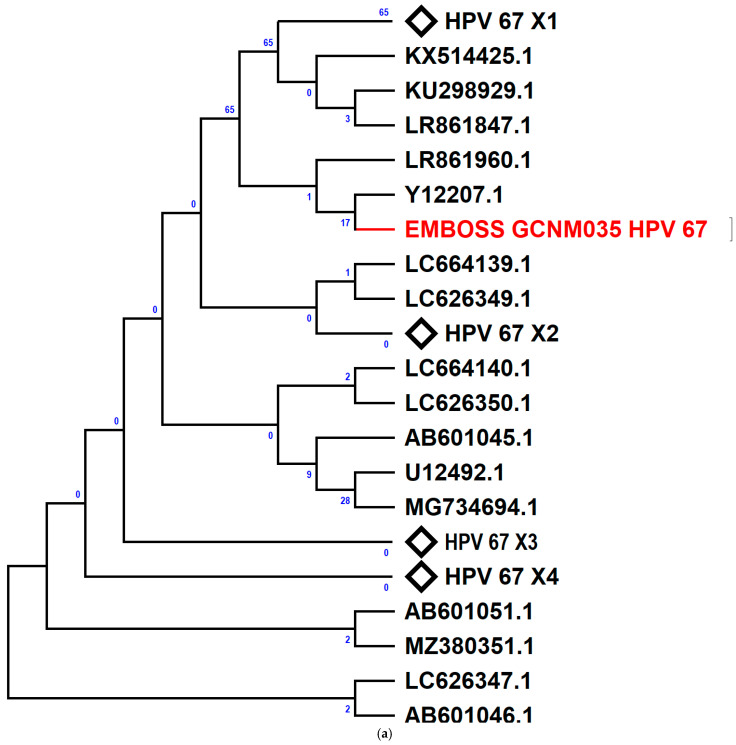
Phylogenetic relationships of HPV 67, HPV 70, and HPV 82 genotypes from this study and reference genomes. The sequences shown in red are from this study. The percentage of replicate trees in which the associated taxa clustered together in the bootstrap test 1000 replicates are shown next to the branches. HPV67 sequences analysis involved 51 nucleotide sequences and a total of 85 positions in the final dataset. HPV 70 sequences analysis involved 36 nucleotide sequences and a total of 96 positions in the final dataset. HPV 80 sequences analysis involved 27 nucleotide sequences and a total of 150 positions in the final dataset. (**a**) showed that the HPV 67 sequence from this study (EMBOSS GCNM035 HPV67) shares strong similarity with a 1997 Italian sequence (Y12207.1). (**b**) showed that the HPV 70 sequences obtained in this study form a distinct cluster, indicating they share a recent common ancestor that is genetically distant from the reference HPV 70 sequences downloaded from NCBI. Notably, two sequences from WLWHIV in this study (EMBOSS GCP 051 et EMBOSS GCP 058) share 100% similarity, suggesting a direct transmission pattern between individuals. These two sequences are also closely related to the other two HPV 70 sequences from this study: EMBOSS GCP083 (WLWHIV) and EMBOSS GCNM049 (WWHIV) (EMBOSS GCP083 et EMBOSS GCNM049). Interestingly, HPV 70 sequences from WLWHIV show higher genetic similarity among themselves compared to those from a woman without HIV (WWHIV). However, all the HPV 70 sequences from this study remain genetically distant from the reference sequences in GenBank. (**c**) showed that the HPV 82 sequence obtained in this study is closely related to a 2013 New York sequence (KF436803.1) downloaded from NCBI [30].

**Table 1 viruses-17-00714-t001:** Demographic characteristics of participants.

Baseline Characteristics	WLWHIV (*n* = 100) (%)	WWHIV (*n* = 51) (%)	*p*-Value
Age (year) 20–29 30–39 40–49 50–59 60–69 No information	11 (11) 21 (21) 38 (38) 24 (24) 6 (6) 0 (0)	11 (21.56) 14 (27.45) 12 (23.53) 10 (19.61) 2 (3.92) 2 (3.92)	*p* = 0.033
Marital status Married Single Widow Divorced	35 (35) 45 (45) 17 (17) 3 (3)	42 (82.3) 8 (15.7) 0 (0) 1 (1.9)	*p* < 0.001
Education level No education Primary Secondary High	26 (26) 29 (29) 40 (40) 5 (5)	1 (1.9) 6 (11.8) 18 (35.3) 26 (51)	*p* < 0.001
Number of sexual partners ≤5 6–10 >10 Not disclosed	77 (77) 6 (6) 3 (3) 14 (14)	23 (45.1) 0 (0) 0 (0) 28 (54.9)	*p* = 0.001
Age at first sexual intercourse <15 15–17 18–20 21–24 >24 No information	3 (3) 30 (30) 52 (52) 7 (7) 3 (3) 5 (5)	3 (5.88) 9 (17.64) 21 (41.17) 6 (11.76) 5 (9.80) 7 (13.72)	*p* = 0.003
Use of Oral contraceptives Yes No No information	38 (38) 54 (54) 8 (8)	17 (33.33) 32 (62.74) 2 (3.92)	*p* = 0.034
Alcohol consumption Yes No No information	81 (81) 19 (19) 0 (0)	12 (23.52) 34 (66.66) 5	*p* < 0.001
Use of condom Always Sometimes Never No information	11 (11) 52 (52) 35 (35) 2 (2)	1 (1.9) 32 (62.74) 16 (31.37) 2 (3.92)	*p* = 0.055
HIV viral load Undetectable <1000 1000–10,000 >10,000 No information	42 (42%) 8 (8%) 14 (14%) 12 (12%) 24 (24%)		
Recent CD4+ counts <200 200–500 >500 No information	6 (6%) 29 (29%) 52 (52%) 13 (13%)		

**Table 2 viruses-17-00714-t002:** Risk factors of any HPV infection in WLWHIV.

Variables	Total	HPV	OR (CI)	*p*-Value	AOR	*p*-Value
Age (years old) ≥35 <35	83 17	69 16	1 3.2 (0.4–26.5)	0.5	-	-
Marital status Married Single	35 65	25 60	1 4.8 (1.5–15.5)	0.01	1 4.6 (1.3–15.9)	0.02
Use of condom Yes No	63 35	53 30	1 1.1 (0.3–3.6)	1.0	-	-
Use of contraception No Yes	54 38	44 34	1 2 (0.6–6.7)	0.4	-	-
Alcohol consumption No Yes	19 81	17 68	1 0.6 (0.13–3.0)	0.7	-	-
Age of first sexual intercourse ≥18 <18	62 33	51 29	1 1.6 (0.5–5.4)	0.6	-	-
Number of sexual partners <4 ≥4	55 31	46 28	1 1.8 (0.4–7.3)	0.5	-	-
Enrollment’s year of ART ≥2014 <2014	48 46	44 36	1 0.3 (0.1–1.1)	0.09	-	-
Initial CD4+ ≥200 <200	41 38	35 33	1 1.1 (0.3–4.1)	1.0	-	-
Frequency of ART use Regular Irregular	29 20	25 16	1 0.6 (0.1–2.9)	0.7	-	-
HIV viral load Undetectable Detectable	53 36	41 34	1 5.0 (1.0–23.7)	0.04	1 5.3 (1.1–26.4)	0.04
Recent CD4+ count ≥500 <500	52 35	42 31	1 1.4 (0.5–6.4)	0.4	-	-

Abbreviations: HPV = human papillomavirus, HR-HPV = high-risk HPV, LR-HPV = low-risk HPV, WLWHIV = women living with HIV, WWHIV = women without HIV, ART = antiretroviral therapy.

**Table 3 viruses-17-00714-t003:** Distribution of any HPV types among both WLWHIV and WWHIV.

HPV Genotypes	Generus + Species	WLWHIV (*n* = 46) (%)	WWHIV (*n* = 21) (%)
High-risk HPV			
HPV 16	α- 9	1 (2.2)	3 (14.3)
HPV 18	α- 7	3 (6.5)	0 (0)
HPV 33	α- 9	4 (8.7)	1 (4.8)
HPV 35	α- 9	0 (0)	1 (4.8)
HPV 39	α- 7	1 (2.2)	0 (0)
HPV 45	α- 7	5 (10.9)	0 (0)
HPV 52	α- 9	2 (4.3)	1 (4.8)
HPV 53	α- 6	1 (2.2)	1 (4.8)
HPV 56	α- 6	1 (2.2)	0 (0)
HPV 58	α- 9	3 (6.5)	0 (0)
HPV 66	α- 6	1 (2.2)	2 (9.5)
HPV 67	α- 9	0 (0)	1 (4.8)
HPV 68	α- 7	0 (0)	1 (4.8)
HPV 70	α- 7	3 (6.5)	1 (4.8)
HPV 82	α- 5	1 (2.2)	0 (0)
Low-risk HPV			
HPV 6	α- 10	2 (4.3)	0 (0)
HPV 32	α- 1	0 (0)	1 (4.8)
HPV 42	α- 1	1 (2.2)	0 (0)
HPV 54	α- 13	3 (6.5)	3 (14.3)
HPV 61	α- 3	0 (0)	1 (4.8)
HPV 62	α- 3	4 (8.7)	2 (9.5)
HPV 72	α- 3	2 (4.3)	0 (0)
HPV 81	α- 3	4 (8.7)	0 (0)
HPV 83	α- 3	3 (6.5)	1 (4.8)
HPV 90		0 (0)	1 (4.8)
HPV 114	α- 3	1 (2.2)	0 (0)

Abbreviations: HPV = human papillomavirus, HR-HPV = high-risk HPV, WLWHIV = women living with HIV, WWHIV = women without HIV.

## Data Availability

All raw data are available as Appendix A.

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
