# Peer review of "HIV–HPV Co-Infection and Identification of Novel High-Risk HPV Among Women at Two Hospital Centers in Cotonou, Republic of Benin"

_viruses, 2025, doi:10.3390/v17050714_

Round 1

Reviewer 1 Report

Comments and Suggestions for Authors

Authors have presented in interesting study about HR-HPV infections in Beninis female population with and without HIV infection. Results are interesting, but the phylogenetic analysis need clarity. Present a table with nucleotide variation obtained from the aligned multi-fasta sequence file of each HPV types submitted as a supplementary file of the manuscript. Show each HPV type eg. HPV16 samples comparisons in table format with the variable site of the dataset. Also present ethnic and likely geographic origin differences of the cases in the manuscript data description and HPV type level distribution.

Authors have presented a manuscript entitled ‘HIV-HPV Co-infection and identification of Novel High-Risk HPV among women at two hospital centers in Cotonou, Republic of Benin´ to be considered for publication in the journal Viruses.   It is an interesting study about HR-HPV infections in Beninis female population with and without HIV infection. Authors present HPV genotype prevalences of a study of 152 women from Benin; 51 without HIV and 100 with HIV infection and treatment. However, the results are not clear nor coherently presented and should be improved before publication.   Main concerns.
  1. Authors should present a study question or hypothesis that has been tested in the study.  
  2. The study is about cervicovaginal HPV prevalences of 152 women from Benin; 51 without HIV and 100 with HIV infection and treatment. The authors should take note from previous publications such as presenting HPV prevalence results between specific sub-populations such as HIV negative and HIV-positive individuals, eg. doi: 10.1016/j.cmi.2015.02.009 study shows in their Table 2 both HIV positive and HIV negative categories and the prevalences are presented organizing the HPV types into their corresponding species groups (between Alpha-2 and Alpha-11). Authors should follow the same method and have two main categories for the prevalence values: HIV-pos and HIV neg and present then in table format, not a bar-plot figure like in the current manuscript version. Then the authors should make a list of all HPV types observed in their dataset and organized them into species groups like in the example article table 2 and present the HPV type level N and prevalence values among HIV positive and among HIV negative groups.
  3.   3. Authors should make a separate table showing the prevalence values of their HIV-neg and HIV Pos groups as single HPV type and multiple HPV type infections. Multiple infections should be shown eg. multiple HPV16, HPV51 and HPV82 infection would be marked in the table as HPV16-51-82 etc. In this table the authors should also calculate the Rischness value and Simpson’s diversity index, example also in Table 2 of the article  doi: 10.1016/j.cmi.2015.02.009.   4. In the discussion section the authors should compare their HPV overall and HPV type level prevalences in HIV-negative women with the HPV prevalence information available for healthy women in different parts of Africa and reported in other articles such as Brune et al 2010 JID doi: 10.1086/657321. Authors should estimate if they have similar or different prevalence values and why as observed in other West African countries.
  4.   5. Phylogenetic analysis per HPV type need clarity. Authors should present a table showing the nucleotide sites that they have sequences from the 151 samples. The table should visualize that how large parto of the L1 gene they actually have sequence data available for each HPV type. This sequence alignment (DNA sequence) should be given as a multi-fasta file as a supplementary file in the article. A phylogeny should be shown only using the sites available from the sampels and it should be clearly reported in the figure legend, which sites of the L1 gene they have used to build the phylogeny.
  Taking together, the authors should conclude in their study if their new sequence based HPV results in HIV-pos and HIV-neg women in Benin shed new light into the HPV type prevalence diversity and distribution in Sub-saharan Africa. Comments on the Quality of English Language

Revise the manuscript language.

Author Response

Dear Reviewer,

Thank you for taking your time to review our manuscript, your comments and suggestions have improved the quality of the manuscript, and we are grateful. Please see the responses to your comments in bold below:

Gouton et al investigate the prevalence of HPV in Benin in women living with or without HIV.  As expected, they find that HPV prevalence is higher in WLWHIV than WWHIV and identify a number of genotypes not previously identified in Benin.

While this study uses a pretty small cohort (100 WLWHIV and 51 WWHIV), it provides some information regarding prevalence beyond previous reports in 2011 and 2016 and there are some statistically significant differences reported.  Bennin has no HPV prevention program and the study provides evidence that such a program is necessary. For this reason, I am reasonably supportive of the manuscript.

Overall, the manuscript is written in very high-quality English. The data analysis seems okay, but the figure quality is probably below the necessary quality to publish. The discussion is probably too long and could use some focusing. There also seem to be some errors and inconsistencies in what is discussed.

Points to consider:

  • Why were only 90 samples of the 116 with detectable HPV sequenced?  This should be explained in the methods. 

Response: 90 samples were selected due to the available resources, and the criteria for selection have been stated in the methods (see lines 187-192). Thank you.

  • There seem to only be two papers on HPV frequence in women in Benin, reference 12 and 37. These should both be cited around line 45 and the mentions of Piras et al on lines 242 and 286 should include the reference number 37 so that it can be easily found in the reference section.  Related to this, it is stated on line 257 that this is the first study in Bennin to look at WLWHIV,yet the introduction (line 46-47) says this was done in their reference 12.  This needs to be consistent and correct. 

Response: We agree with this observation and have corrected it in the revised manuscript. Thank you.

  • Figure quality is low and poorly organized stylistically. In Fig. 1, Panel A is more or less okay.  Panels B and C should be much smaller and font sizes and types in the panels should be consistent. In figure 2, the graphic "trees" are far too large and the text is far too small.  The figure seems to be of low resolution as well, being fuzzy and pixilated. This needs work to look professional, instead of looking somewhat amateurish.

Response: This is noted, we have now converted figure 1 to table 3 to capture other details not clear with the bar chart, thank you.

  • For Table 1, some rows have 0 (0) listed and others are left blank that are 0.  This should be consistent.  For Table 2, what is AOR?  I assume its adjusted odds ratio, but this is not defined and no description of the adjustment is provided as far as I noticed.

Response: Noted and corrected, thank you.

  • Some of the discussion if not relevant.  The comparisons with Pakistan (a country) and Tehran (a city) seem totally irrelevant on lines 285-290. The comparison with nearby countries is import, as for Ghana and Nigeria (lines 252-260), but they should state whether those countries have HPV vaccine programs (which Benin does not) and try to tease out whether the lack of a vaccine program in Benin is visible in higher HPV incidence.  This would be fuel to use in trying to campaign for an HPV vaccine program.

Response: Noted and corrected, thank you.

6) The discussion on HPV vaccine protection overlap with detected HPV types is a good start, but they might consider how "cross-protection" across closely related HPV types, which we know quite a lot about and how this may or may not be relevant to the types observed in Benin.

Response: Noted and corrected, thank you.

Overall, we are grateful for your comments. Thank you.

Reviewer 2 Report

Comments and Suggestions for Authors

Abstract

You should explain you novel finding shortly but you explain like as your result. What is still unknow in your research no mentioned ?

Introduction

There is no explanation about the diseases. How it occurs and why it is important ? 

method 

Study Design and Population is not sufficient. Please explain briefly. 

In the bioinformatics analysis, you just mentioned the software name, no in details. Please explain everything how you analysis and what is the parameter. 

The sample number is low. I think this is not enough. Is it possible to increase ?

Result

Please follow the rules of making Table.

You should use line otherwise it is difficult to understand. 

The most important thing, the result parts have no paragraph with title. Its look like massy. Proper title of each find is important to understand the novelty of your data.

please prepare the result according to the journal style.

Fig 1 the graph is not acceptable. no error bar, no statistical analysis. 

The image quality is very poor. Use Graph Pad prism. 

Phylogenetic data is too strange. Why no proper explanation ?

You just got the data from NCBI, please compare the group and make a story. 

Image again very poor. 

Discussion need to organize more clearly. 

Please explain logically and with evidence. You may present look like result work. 

please modify it. 

Another important thing, when you explain any data, please tell the possible reason for that. 

In the discussion part, you should tell at least two possible hypothesis about your data. 

Where is the conclusion ? 

Comments on the Quality of English Language

ok

Author Response

Dear Reviewer,

We are grateful that you found our manuscript worthy of your assessment and time. Your comments and suggestions have helped us improve the quality of the manuscript. Thank you. Please, see below in bold our responses to the comments and suggestions:

Abstract

You should explain you novel finding shortly but you explain like as your result. What is still unknow in your research no mentioned ?

Response: We mentioned that the prevalence and genotype distribution are unknown in Benin in line 17 of the abstract. This is what is unknown and the objective of our study, thank you.

Introduction

There is no explanation about the diseases. How it occurs and why it is important? 

Response: Thank you for this comment, we have now revised the introduction section to include your suggestion, thank you

Method 

Study Design and Population is not sufficient. Please explain briefly. 

Response: We have explained the rationale behind our study design in the method section (2.1)  and have now added the statistical basis for sample selection (Line 114-124). Thank you.

In the bioinformatics analysis, you just mentioned the software name, no in details. Please explain everything how you analysis and what is the parameter. 

Response: We have now updated and revised this to reflect the comments, thank you.

The sample number is low. I think this is not enough. Is it possible to increase ?

Result: The sample size was calculated on a statistical basis using reported HPV prevalence among women living with HIV in Cotonou, Benin. Details are provided in the manuscript. Unfortunately, we will not be able to increase the sample size because the duration and ethical approval for the study have elapsed. Thank you.

Please follow the rules of making Table.

Response: This is noted and corrected

You should use line otherwise it is difficult to understand. 

Response: We have updated the tables as requested. Thank you.

The most important thing, the result parts have no paragraph with title. Its look like massy. Proper title of each find is important to understand the novelty of your data.

Response: We used the journal style, which does not require us to use subheadings or subtitles in the discussion, thank you

please prepare the result according to the journal style.

Response: This is what we did, thank you.

Fig 1 the graph is not acceptable. no error bar, no statistical analysis. 

Response: Thank you we have now updated this in the revised figure 1.

The image quality is very poor. Use Graph Pad prism. 

Response: We used Graph Pad Prism and have improved the quality of the figure, thank you.

Phylogenetic data is too strange. Why no proper explanation ?

You just got the data from NCBI, please compare the group and make a story. 

Response: We have added an explanation and discussed accordingly. Thank you

Image again very poor. 

Response: This has been improved, thank you

Discussion need to organize more clearly. 

Response: This has been improved, thank you

Please explain logically and with evidence. You may present look like result work. 

please modify it. 

Response: We have improved the discussion, thank you

Another important thing, when you explain any data, please tell the possible reason for that.

Response: We have done this, thank you 

In the discussion part, you should tell at least two possible hypothesis about your data. 

Response: Thank you for the comment. We have mentioned the overall hypothesis for the study in the introduction. We discussed our data in context with what is already published and suggested a possible explanation for the agreement or differences with our results. Discussion style varies, and we are not required or limited to the use of a hypothesis to discuss our results. Thank you.

Response: Where is the conclusion ? 

Response: The conclusion can be found from lines 489-496. Thank you.

Again, thank you for reviewing our manuscript. Your comments have improved the quality of the revised version.

Reviewer 3 Report

Comments and Suggestions for Authors

Gouton et al investigate the prevalence of HPV in Benin in women living with or without HIV.  As expected, they find that HPV prevalence is higher in WLWHIV than WWHIV and identify a number of genotypes not previously identified in Benin.

While this study uses a pretty small cohort (100 WLWHIV and 51 WWHIV), it provides some information regarding prevalence beyond previous reports in 2011 and 2016 and there are some statistically significant differences reported.  Bennin has no HPV prevention program and the study provides evidence that such a program is necessary. For this reason, I am reasonably supportive of the manuscript.

Overall, the manuscript is written in very high-quality English. The data analysis seems okay, but the figure quality is probably below the necessary quality to publish. The discussion is probably too long and could use some focusing. There also seem to be some errors and inconsistencies in what is discussed.

Points to consider:

1)  Why were only 90 samples of the 116 with detectable HPV sequenced?  This should be explained in the methods. 

2) There seem to only be two papers on HPV frequence in women in Benin, reference 12 and 37. These should both be cited around line 45 and the mentions of Piras et al on lines 242 and 286 should include the reference number 37 so that it can be easily found in the reference section.  Related to this, it is stated on line 257 that this is the first study in Bennin to look at WLWHIV,yet the introduction (line 46-47) says this was done in their reference 12.  This needs to be consistent and correct. 

3) Figure quality is low and poorly organized stylistically. In Fig. 1, Panel A is more or less okay.  Panels B and C should be much smaller and font sizes and types in the panels should be consistent. In figure 2, the graphic "trees" are far too large and the text is far too small.  The figure seems to be of low resolution as well, being fuzzy and pixilated. This needs work to look professional, instead of looking somewhat amateurish.

4) For Table 1, some rows have 0 (0) listed and others are left blank that are 0.  This should be consistent.  For Table 2, what is AOR?  I assume its adjusted odds ratio, but this is not defined and no description of the adjustment is provided as far as I noticed.

5) Some of the discussion if not relevant.  The comparisons with Pakistan (a country) and Tehran (a city) seem totally irrelevant on lines 285-290. The comparison with nearby countries is import, as for Ghana and Nigeria (lines 252-260), but they should state whether those countries have HPV vaccine programs (which Benin does not) and try to tease out whether the lack of a vaccine program in Benin is visible in higher HPV incidence.  This would be fuel to use in trying to campaign for an HPV vaccine program.

6) The discussion on HPV vaccine protection overlap with detected HPV types is a good start, but they might consider how "cross-protection" across closely related HPV types, which we know quite a lot about and how this may or may not be relevant to the types observed in Benin.

Author Response

Dear Reviewer,

We appreciate your time and comments on our manuscript, please see below the response to the comments and the revised and improved manuscript:

Authors have presented in interesting study about HR-HPV infections in Beninis female population with and without HIV infection. Results are interesting, but the phylogenetic analysis need clarity. Present a table with nucleotide variation obtained from the aligned multi-fasta sequence file of each HPV types submitted as a supplementary file of the manuscript. Show each HPV type eg. HPV16 samples comparisons in table format with the variable site of the dataset. Also present ethnic and likely geographic origin differences of the cases in the manuscript data description and HPV type level distribution.

Response: Thank you for the suggestion. We have corrected this and attached the supplementary file as requested.

Authors have presented a manuscript entitled ‘HIV-HPV Co-infection and identification of Novel High-Risk HPV among women at two hospital centers in Cotonou, Republic of Benin´ to be considered for publication in the journal Viruses.   It is an interesting study about HR-HPV infections in Beninis female population with and without HIV infection. Authors present HPV genotype prevalences of a study of 152 women from Benin; 51 without HIV and 100 with HIV infection and treatment. However, the results are not clear nor coherently presented and should be improved before publication.   Main concerns.

  1. Authors should present a study question or hypothesis that has been tested in the study.  

Response: Thank you for the suggestion, we have included the hypothesis on Line 91 as suggested.

  1. The study is about cervicovaginal HPV prevalences of 152 women from Benin; 51 without HIV and 100 with HIV infection and treatment. The authors should take note from previous publications such as presenting HPV prevalence results between specific sub-populations such as HIV negative and HIV-positive individuals, eg. doi: 10.1016/j.cmi.2015.02.009 study shows in their Table 2 both HIV positive and HIV negative categories and the prevalences are presented organizing the HPV types into their corresponding species groups (between Alpha-2 and Alpha-11). Authors should follow the same method and have two main categories for the prevalence values: HIV-pos and HIV neg and present then in table format, not a bar-plot figure like in the current manuscript version. Then the authors should make a list of all HPV types observed in their dataset and organized them into species groups like in the example article table 2 and present the HPV type level N and prevalence values among HIV positive and among HIV negative groups.
  2.  

Response: Thank you, this has now been addressed. Figure 1a has now been converted to Table 3 to capture the points highlighted.

  1.   3. Authors should make a separate table showing the prevalence values of their HIV-neg and HIV Pos groups as single HPV type and multiple HPV type infections. Multiple infections should be shown eg. multiple HPV16, HPV51 and HPV82 infection would be marked in the table as HPV16-51-82 etc. In this table the authors should also calculate the Rischness value and Simpson’s diversity index, example also in Table 2 of the article  doi: 10.1016/j.cmi.2015.02.009.  

Response: The assay used (Conventional PCR and sequencing) in our study only detected a single HPV infection at a time. We were not able to detect HPV-coinfection or multiple infections. Thank you.

  1.  4. In the discussion section the authors should compare their HPV overall and HPV type level prevalences in HIV-negative women with the HPV prevalence information available for healthy women in different parts of Africa and reported in other articles such as Brune et al 2010 JID doi: 10.1086/657321. Authors should estimate if they have similar or different prevalence values and why as observed in other West African countries.

Response: Thank you for this important observation. We have included this in the revised discussion.

  1.   5. Phylogenetic analysis per HPV type need clarity. Authors should present a table showing the nucleotide sites that they have sequences from the 151 samples. The table should visualize that how large parto of the L1 gene they actually have sequence data available for each HPV type. This sequence alignment (DNA sequence) should be given as a multi-fasta file as a supplementary file in the article. A phylogeny should be shown only using the sites available from the sampels and it should be clearly reported in the figure legend, which sites of the L1 gene they have used to build the phylogeny.

Response: We agree with you, we have included the tables in supplementary files (S3-S5) and have uploaded the multi fasta files of the sequence alignment as requested. Thank you.

  Taking together, the authors should conclude in their study if their new sequence based HPV results in HIV-pos and HIV-neg women in Benin shed new light into the HPV type prevalence diversity and distribution in Sub-saharan Africa.

Response: This is noted and corrected in the revised discussion. Thank you.

Comments on the Quality of English Language

Revise the manuscript language.

Response: We have improved the grammar, thank you.

Once again, thank you for your time and comments.

Round 2

Reviewer 2 Report

Comments and Suggestions for Authors

This is very basic to learn how to present the data.

In your case you show very big in one pages, but why ?

Did you check other paper, how they present data in paper ?

If no one read you phylogenetic tree, what is the importance ? As I mentioned this is not readable. 

Please improve first. 

Author Response

We have now revised and replaced the trees with improved versions (Line 291- 323). We explained the tree construction in lines 173 to 183 and discussed it in lines 403-420. Thank you.